# Clinical Practice Guidelines: An Opinion of the Legal Implication to Veterinary Medicine

**DOI:** 10.3390/ani9080577

**Published:** 2019-08-19

**Authors:** Michela Pugliese, Eva Voslarova, Vito Biondi, Annamaria Passantino

**Affiliations:** 1Department of Veterinary Sciences, University of Messina, Polo Universitario Annunziata, 98168 Messina, Italy; 2Department of Animal Protection, Welfare and Behaviour, Faculty of Veterinary Hygiene and Ecology, University of Veterinary and Pharmaceutical Sciences Brno, 612 42 Brno, Czech Republic

**Keywords:** clinical practice guidelines, standard of care, veterinary profession, veterinary malpractice, law

## Abstract

**Simple Summary:**

With the changing nature of the bond between humans and animals over the past decades, society has higher expectations for veterinary profession services and considers damages in veterinary malpractice and liability cases more carefully. In veterinary malpractice litigation, standards of care expressed in guideline statements could influence the civil and penal courts in the decision-making process. Based on these considerations, the authors examine the importance of clinical practice guidelines (CPGs) in veterinary malpractice litigation involving quality of care and explore how the law may treat CPGs in the future.

**Abstract:**

The strengthening of the bond between humans and animals has changed the landscape of the veterinary profession. This has, in turn, led the legal system to assess damages in veterinary malpractice and liability cases more carefully, paying attention to the possibility of using clinical practice guidelines (CPGs) to prove whether the defendant veterinarian contravened or not the standard of care. In this era of evidence-based veterinary medicine, CPGs are becoming an integral part of many aspects of veterinary practice, even if CPGs do not have the force of law and are situated halfway between ethical rules and legal requirements. Although guidelines have been used for several years, there seems to be a general lack of recognition of the medical and legal ramifications of CPGs for veterinarians. This creates ambiguity and inconsistency in the care that veterinary practitioners provide, compromises the care animals receive, and prevents the courts from assessing veterinarian competence in a systematic and rational way. On the basis of these considerations, this article discusses the legal implications of CPGs in veterinary medicine for dogs and cats and explores how the law may treat CPGs in the future. Redefining the CPGs should be a priority for veterinary profession. NOTE: The authors chose to use the terms “companion animal,” “pet,” and “small animal” interchangeably throughout this article, as all three are commonly in use and refer to the same animals (dogs and cats).

## 1. Introduction

The veterinary companion animal practice can be compared in many aspects with that of human medicine [1] because it has specialized practice areas (such as neurology, cardiology, oncology, ophthalmology, dermatology, etc.) and a wide range of preventive care, as well as treatments for major diseases.

The type of medical care provided to animals is as sophisticated as the care available to humans: The companion animals that are brought into a home environment receive kidney/liver transplants, hemodialysis, chemotherapy to treat cancer, pain medication, etc. [2,3,4,5,6,7,8,9,10,11,12].

The development of knowledge in veterinary medicine has determined a change of veterinary practice focusing on the health of the animal-patient and its best interests in consideration of the growing recognition of the important human-animal bond. Although animals have traditionally been viewed as mere property [13,14,15], today animal guardians consider their pets as members of the family, children, or best friends rather than as personal property and describe the animal’s role in the family as “very important” [16]. Animals, and especially companion animals, walk a fine line between owned properties and being members of a family. In recent decades, the status of animals has also changed in the legislation of many European countries. 

The stages of this evolution are marked by some important documents. The Universal Declaration of Animals’ Rights, proclaimed on October 15, 1978 at UNESCO (United Nations Educational, Scientific and Cultural Organization) House in Paris, recognizes that animals have rights and establishes that the violation of such rights lead man to commit crimes against the nature. Notably, it asserts that there cannot be respect among men if, at first, they do not respect animals [15]. The Declaration represents the starting point for all the events that have taken place since, like the European Convention for the Protection of Companion Animals [17] that recognizes the man to have the moral obligation to respect all living creatures, also “in consideration of the particular ties existing between man and companion animals”. The European Convention was implemented in Italy in 2010 [18], but many of its precepts have already been acknowledged by Law no. 281 of August 14, 1991 [19]. This, at last, shows a radical change of perspective in juridical guardianship, with the awareness of the fact that an animal is a psychophysical entity capable, like man, of feelings and emotions, pain, and anguish. The animal is considered as a subject with rights, and so fully to be safeguarded. It is no longer an object regarded only as “res” useful to man. The Article 1 of the aforesaid law indicates the state as the fundamental promoter of such guardianship. Therefore, the “Safeguarding of Animal Welfare” aims to recognize animals’ role and habitat, considering them as our fellow earthly tenants, reducing their exploitation and subjection by man.

It must be specified that this concept is a part of a wider movement at a community level. In fact, the Article 13 of the Treaty on the Functioning of the European Union (TFEU) [20] obliges the European Union and its Member States to take into account regard to the welfare requirements of animals in that they are sentient beings. The recognition of animal dignity as sentient beings, contained in the EC Treaty’s Protocol on Protection and Welfare of Animals, demonstrates how strongly the need for animal safeguard and welfare is perceived by the EU Members [21] and constitutes a value strongly shared by most European citizens, including Italians.

There is a growing group of countries where the respect for animals is also recognized as a legal value, which allows animals to benefit from specific legal protection.

A great example is Germany that implemented the protection of animals (considered as legal creatures) in its federal Constitution (Deutscher Bundestag, Article 20a) [22] in 2002, becoming one of the first European Union Member States to do so. Its Civil Code, article 90a [23], has established a negative definition of animal as not being objects.

Since 2002 in Switzerland in the Zivilgesetzbuch “Tiere keine Sachen mehr” (Art. 641a Abs. 1 ZGB) [24], in Article 285a of the Austrian Civil Code (Allgemeines Bürgerliches Gesetzbuch) (ABGB) [25], which became operational in 1988, ‘animals are not objects, but they are protected by special laws’. Similarly, in Article 511-1 of Civil Code of Catalonia (Código Civil de Cataluña) [26] the animals, which are not considered as objects, are under the special protection by the laws. 

The Civil Code of the Czech Republic, no. 89/2012, Article 494 states that “Living animals have special importance and value as living creatures endowed by senses. Living animals are not objects; provisions regarding objects are to be applied to animals only if this application does not contravene with the nature of the animal” [27]. 

There are other countries that have made this recognition in civil law as France (article 515.14) [28], Quebec (Article 898.1) [29], Colombia (Article 655 and Law 1774 of 6.01.2016) [30], which modifies the Civil Law.

In Italy, although animals’ suffering is recognized by the law [31], which aims to prevent it by making certain behavior obligatory, animals are still juridically considered as “objects” and as goods owned by man (Articles 810 and 812 of the Italian Civil Code) [14].

These changes occurred as a consequence of the growing citizen’s demand for laws to provide better protection for all animals, taking into account that society is ever more sensitive to the animal issue. 

Given the changing nature of the bond between humans and companion animals and the practice of veterinary medicine, the public has higher expectations for veterinary profession services now than in previous decades and people are willing to spend more money on medical care.

Certainly, there are owners who are willing to spend a significant amount of money on sophisticated procedures in order to save their companion animal’s life.

Consequently, the legal system should assess damages caused by veterinary malpractice and liability cases [32] (It is difficult to obtain statistics on the number of cases brought against veterinarians for malpractice. Anyway, the public’s perception is that there is a litigation explosion resulting in the increasing the calls received by lawyers. The effect is to make people think that if anything goes wrong, they can get significant compensation, presumably increasing the total number of claims. Nevertheless, Soave (2000) [33] states that there are more than 2,000 cases of malpractice and negligence filed in U.S. courts each year) more carefully than in the past because, as described above, animals are considered sentient beings, paying attention to the possibility of using clinical practice guidelines (CPGs) to prove whether the defendant veterinarian breached the standard of care.

In the era of evidence-based veterinary medicine [34], CPGs are becoming an integral part of many aspects of small animal practice. CPGs regarding cardiology [35], dermatology [36], oncology [37], nephrology [38], and antimicrobial treatment [39] have been implemented (see paragraph 3). They are increasingly credited with pivotal significance in highly diverse areas of human conduct in order to improve clinical practice not only in human medicine, but also in veterinary medicine. In this context, CPGs gain a new cultural ascendancy. 

Despite their popularity, CPGs are not used often in clinical practice and their use remains controversial [40]. Different problems are correlated to applicability of CPGs, including difficult accessibility, long lifecycle of CPG advancement, inappropriateness to local situations, and lack of active user engrossment [41]. The majority of CPGs are based on trials considering homogenous populations. In veterinary medicine this is a limiting factor in their application, because in clinical practice, patients are rarely homogenous (i.e., species, breed, body weight, indoor/outdoor, etc.). Therefore, the active involvement of professional and clinician users in drafting is necessary to prepare the guidelines directed at the disease and not at a particular patient [42].

In deciding actions in negligence, courts could be influenced by standards of care expressed in guideline statements. 

This article begins with a description of CPGs followed by their use in the court. Finally, we discuss the legal implications of CPGs in veterinary medicine and explore how the law may treat CPGs in the future. Redefining the CPGs should be a priority for veterinary profession.

## 2. The Changing Nature of Veterinary Medicine

Over the past two decades with increasing urbanization, the strengthening of the bond between humans and animals has changed the landscape of the veterinary profession [15]. Given that, nowadays, most households own a pet and many pets are just as important as a family member or friend, sometimes more, the owners invest in health of their animals. Consequently, demand for veterinary medical services (e.g., cost of a treatment to improve the animal’s quality life) has grown significantly similar to that of human medicine.

Since veterinary medicine has developed a wide range of highly qualified technologies, such as computed axial tomography, magnetic resonance imaging [43,44], etc., and treatment options, the veterinarian has the opportunity to provide wider services and more high-quality care to the animal patients, increasing owner’s expectations in this manner.

In view of scientific progress in veterinary medicine and the recognizing the animals as sentient beings in the European law [14,15,20], the standard of animal care has changed and risen. 

As a result, veterinarians must practice a superior quality of medicine.

The clinical decisions must be not made on the basis of what has been used previously by an individual clinician or, in unusual clinical cases, according to advice from colleagues, specialists, laboratories, or the internet. Instead, clinical decisions should be made by following the CPGs.

## 3. Overview of Clinical Practice Guidelines (CPGs)

The Institute of Medicine (1990) [45] defines the CPGs as “systematically developed statements to assist practitioner and patient decisions about appropriate care for specific clinical circumstances”. They can be considered a written statement describing the best clinic practices by applying to animal-patient care based on the professional judgment of a given group of medical professionals (developers) in a given practice area. Taking into consideration that CPGs originate from the consensus of experts, they are considered as a prevalent standard of care in the veterinary profession and designed to improve the decision-making process. 

They are situated halfway between ethical rules, code of conduct, and legal requirements. In fact, they are inspired by different principles (i.e., principle of respect for animal rights and dignity, of justice, of beneficence, of non-maleficence, of science and conscience, of continuing professional development) that underlie medical science, considered as an expression of diligence of the veterinarian.

Practice protocols and standards are terms often used, or nearly so, as synonymous with the concept guidelines, but they are different. In fact, the first is a framework that outlines the offered care to animal-patient (why, where and when, and by whom the care is given) in a well-defined area of practice. The second, applied as an audit, is a consensus statement that recognizes the expected results. 

CPGs can be promulgated by a variety of organizations (public and/or private) that have credibility with respect to the veterinary profession (district health authorities, hospital departments, professional associations, general practitioners, and scientific societies) (Table 1). 

Among the different and professed aims to which the CPGs tend, there are: (1) Improving healthcare quality; (2) decreasing healthcare costs using review criteria; (3) promoting appropriate use of medical technologies; (4) improving animal owner awareness of healthcare needs; (5) reducing the risks of legal liability in healthcare delivery; (6) incorporating research findings into medical practice [46].

CPGs are commonly couched in language, which assumes a tone of authority, displaying a gradient of exhortation from mere option to explicit moral imperative i.e., utilizing the following words: “may” (See in Step 3: Develop a Personalized EOL Treatment Plan of AAHA/IAAHPC End-of-Life Care Guidelines: (…) For example, performing radical surgery requiring a significant amount of rehabilitation may not be in the patient’s best interest if expected survival time is short.); “should” (See for example AAHA/IAAHPC End-of-Life Care Guidelines: The veterinarian should advise the client about the expected trajectory of the pet’s disease. This should include a discussion of diagnostic and treatment options.); or “must” (See in Step 4: Implement Palliative or Hospice Care of AAHA/IAAHPC End-of-Life Care Guidelines: (…) In order for EOL care to be successful, not only must the client be willing and able to implement the treatment plan, the patient must also be a willing participant. This is analogous to delivery of medical care to children.) in order to follow the proffered advice. 

Guidelines developed or adopted by prestigious organizations may gain wide influence by virtue of the approval which they carry as a result.

In medicine, according to the Institute of Medicine (1992) [47], CPGs should possess the following characteristics: (i) Validity where guidelines are regarded as valid if the expected results are achieved by following the recommendations; (ii) reliability in which the guidelines are interpreted and applied in a similar fashion and in a systematic and rigorous manner by those who intend to follow them; (iii) clinical applicability and flexibility require that the target populations to which the guidelines apply be defined conforming to scientific evidence and that guidelines be flexible by identifying exceptions and how patient preferences are to be taken into account in the decision making process; (iv) clarity in which the guidelines may be written to be technically accurate but also easy to understand; (v) multidisciplinary process, where an multidisciplinary approach depending on the complex coordination of many factors and efforts of many people (representatives of key affected groups and disciplines) could prove the implementation of a quality improvement; (vi) scheduled review, meaning that the guidelines should be periodically reviewed to integrate, e.g., new knowledge; and (vii) documentation, wherein the guideline development process should be public and thoroughly documented. 

### The Standard of Care (SOC)

To understand how CPGs might or might not be appropriate decision tools in malpractice lawsuits, it is important to know what the standard of care (SOC) is. 

The standard is difficult to define, because it is usually based on what a “reasonable and prudent” physician working under similar conditions would do. 

Block [63] reports that the SOC is defined as *“the standard of care required of and practiced by the average reasonably prudent, competent veterinarian in the community. (…) nor does the legal standard set the threshold for liability at a particularly high level. The average or normal practitioner, not the best or most highly skilled, sets the standard”*.

In legal terms, this means that SOC indicates the degree of care and skill of the average, prudent veterinary provider in a given community, taking into account the medical knowledge that is available to the veterinarian. It is how similarly qualified practitioners would have managed the patient’s care in the same or similar circumstances.

SOC varies from state to state. Some states use the “locality rule” [64] standard of care that provides veterinarians are held to the same standards as other veterinarians in their geographical area. 

The locality rule evaluates the conduct of a professional by considering the professional standards in the geographical area where the professional practices. The geographical area may be as narrow as the immediate locality or as large as a national standard. 

Just as with the medical profession, with the increasing access to information and continuing professional education requirements, it appears the veterinary profession should adopt a more uniform standard [65]. A more general uniform standard promotes higher levels of competence within the profession [65]. Notwithstanding any applicable geographical limitation, veterinarians who hold themselves as specialists in a particular aspect of veterinary practice should be held to the standards of other specialists in that field [64]. 

Given the increased availability of specialists, a veterinarian can also be liable if he/she fails to refer a client to a specialist in appropriate circumstances. 

## 4. Use of CPGs in the Court

Although CPGs produced do not have the force of law, they function much like expert testimony to inform the court about the nature of existing practice. As will be discussed below, this use “judicial” of CPGs seems likely to be accepted by the courts. In any case, when considering testimony supported by CPGs, courts assess some factors such as the case’s type, the source of the guideline, the expert’s own acknowledgment of its relevance and reliability, and whether the expert’s testimony itself is reliable.

To prove a claim for veterinary malpractice, despite the laws of medical malpractice varying significantly from nation to nation, an owner plaintiff (the person bringing the action) must be able to show that: (1) The defendant veterinarian owed the plaintiff a duty of care in order to prevent any injuries caused to the animal; (2) the veterinarian breached that duty of care by failing; and (3) the breach of duty has led to a damage the plaintiff, a damage that should have been foreseeable and reasonably avoidable. 

Actions that may constitute malpractice include, for example, misdiagnosis, incorrect treatment, failure of skill in surgical or nonsurgical procedure, violation of commonly acceptable protocols, failure to provide preventative care, etc. 

(See Daughen v. Fox, 539 A.2d 858 (Pa. Super. 1988) Plaintiffs brought a claim for intentional infliction of emotional distress and loss of companionship after defendant animal hospital performed unnecessary surgery based on a mix-up of x-rays.

DeLany v. Kriger, Slip Copy, 2019 WL 1307453 (Tenn. Ct. App. Mar. 20, 2019). This case concerns a veterinary negligence action. The owners of a cat filed a wrongful death complaint against the cat’s veterinarian and animal hospital after the cat was killed when the veterinarian wrongly placing a feeding tube into the cat’s trachea rather than her esophagus, causing the cat to aspirate and die when she was fed through the tube. 

Jack W. Dyess v. Hugh L. Caraway, 190 So.2d (666 La. App., 1966). Plaintiff claimed damages for the death of five pedigreed Norwegian Elkhound puppies resulting from the negligence of defendant, a duly licensed veterinarian. Specifically, defendant allegedly failed to make proper diagnostic tests, failed to give proper treatment for coccidia from which the puppy died although the defendant had professional knowledge that the puppy was suffering from that disease, and failed to exercise the standard of care required by the average prudent veterinarian in the community.

Bradley Gilman, DVM, Appellant, v. Nevada State Board of Veterinary Medical Examiners, Respondent, 89 P.3d 1000 (Nev. 2004). The Slensky’s took their ill beagle to Defendant’s Animal Hospital for routine vaccinations and examinations due to the dog having loose stools for four days. X-rays of the dog were taken, and when the dog was returned to the Slensky’s, it collapsed. Defendant instructed them to take the dog to the emergency clinic, where it later died. The family filed a complaint with the Nevada State Board of Veterinary Medical Examiners, and defendant was later convicted of gross negligence and incompetence, an ethics violation, and for using an unlicensed veterinary technician.

Gonzalez v. South Texas Veterinary Associates, Inc., 2013 WL 6729873 (Tex. App. Dec. 19, 2013). Plaintiff acquired an indoor/outdoor cat with an unknown medical and vaccination history. Plaintiff took cat to defendant for treatment and the cat received a vaccination. The cat soon developed a golf-ball-sized mass that contained a quarter-sized ulceration which was draining “matter” on the cat’s right rear leg. When plaintiff returned the cat to the defendant, defendant diagnosed the cat with an infection, prescribed an antibiotic for treatment, and instructed Gonzalez to return if the cat’s symptoms did not improve. When the cat’s symptoms did not improve, plaintiff took the cat to another veterinarian, who diagnosed the cat with vaccine-associated sarcoma. The cat eventually had to be euthanized. Acting pro se, the plaintiff filed suit, alleging that defendant failed to: (1) Inform her of vaccine-associated sarcoma risk; (2) adhere to feline vaccination protocols; and (3) properly diagnose vaccine-associated sarcoma in the cat, which resulted in the loss of her life.

Johnson v. Wander, 592 So. 2d. 1225 (Fla. Dist. Ct. App. 1992). Petitioner pet owner alleged that respondent veterinarian took her dog to be spayed and left the animal on heating pads which resulted in serious burns, so petitioner filed a claim for damages on the basis of gross negligence, damage to property, and emotional distress. The trial court entered partial summary judgments on the claims for punitive damages and emotional distress and, on a subsequent motion, transferred the case to the county court as a claim for less than the circuit court jurisdictional amount. The appellate court held that there remained a jury question on the issues of gross negligence and physical and mental pain and suffering as claimed by petitioner.

Shera v. N.C. State University Veterinary Teaching Hosp., 723 S.E.2d 352 (N.C. Ct. App. 2012). After an animal hospital caused the death of a dog due to an improperly placed feeding tube, the dog owners sued for veterinary malpractice under the Tort Claims Act. The Court of Appeals held that the replacement value of the dog was the appropriate measure of damages, and not the intrinsic value. Owners’ emotional bond with the dog was not compensable under North Carolina law).

When a veterinarian agrees to treat an animal, he/she owes to their clients (owner and animal patient) the duty to practice a veterinary medicine based on a professional morality which consists of a strict adherence to an overall standard of ethical behavior befitting the dignity and integrity of the profession whose responsibility it is to provide animals with the highest possible standard of medical care.

For instance, suppose Veterinarian X neuters Client Y’s dog by removing both testicles. After the surgery, the animal dies as a result of a fatal hemorrhage due to failure to ligate the blood vessels by the practitioner.

Veterinarian X may argue that other veterinarians in the geographical community neuter dogs with a similar procedure.

Whether Veterinarian X owed a duty to provide a better method of hemostasis depends on the standard of care that would be applied in the case.

The failing duty to exercise reasonable care in order to ensure the safety of other humans may be another example. In practice, a veterinarian may be a subject to negligence claim if a dog injures (i.e., bites) his/her owner during the physical examination because the animal was agitated.

In this case, the veterinarian is liable for failure to uphold his/her duty to prevent the bite by having a trained professional restrain the dog, even though the animal belonged to the owner and even though the same owner expressed a desire to restrain the dog during the visit. 

Indeed, duty of care places a moral duty to anticipate possible causes of injures and to do everything reasonably practicable to mitigate these possible causes.

In veterinary medicine, courts have recently begun to use CGPs to address the SOC in medical practice litigation.

In lawsuits, both animal owners (plaintiffs) and veterinarians (defendants) can apply CGPs and SOC, e.g., for inculpatory (blame-placing) purposes when an animal owner asserts that the defendant should have followed an appropriate guideline but did not. 

In theory, the plaintiff proves that his/her animal was injured because the veterinarian failed to meet a SOC expected in the community and there is a causal relationship between the injury and the failure to meet the SOC (negligence). In the same manner, the veterinarian could apply a CPG for exculpatory (blame-relieving) purposes in order to show that he/she operated in accordance with an applicable guideline. For instance, a veterinarian could take note of a guideline and decide not to perform a certain diagnostic test. The animal owner may later allege that in omitting the test, the veterinary practitioner failed to make an important diagnosis and, as a result, the animal patient suffered an injury. Supposing the veterinarian is simply trying to provide appropriate care, he/she must adapt any use of a guideline’s recommendations to the individual animal. If this is done in a reasonable manner, it stands to reason that the court will find the guideline itself as persuasive evidence that the veterinarian has met a standard of care.

In either case, the party asserting the guideline is asking the court to accept it as a proof that the veterinarian either did (exculpatory use) or did not (inculpatory use) meet the legally required standard of care. 

Although an expert witness as evidence of accepted SOC could present CGPs to a court, they cannot replace the expert testimony. Courts are unlikely to adopt SOC advocated in clinical guidelines as legal “gold standards” because the mere fact that a guideline exists does not itself establish that compliance with it is reasonable in the circumstances or that noncompliance is negligent. 

To evaluate the possible liability of the veterinarian, the expert witness is generally called to answer to following topics: (1) The applicable SOC [63]; (2) causation (the association between the supposed unlawful conduct and the damage suffered by the plaintiff); and (3) the assessment of damages, which often involves (4) medical prognosis. 

Given that courts usually evaluate the adequacy of an animal’s treatment by its conformity to standard practice in the relevant medical community, relating to the points (2), (3), and (4), the expert witness must apply his/her knowledge directly to the question at issue and testify what other veterinarians would usually do in a similar clinical situation. 

The distinction concerning the SOC questions is essential because a court adopts a CPG to determine the legal SOC, making the presence of expert testimony about the dispute unnecessary. Contrarily, even in the event that a CPG presents significant data regarding to the causation, estimation of damages, or prognosis, it is difficult for a court to value the facts without the deposition of medical expert witness. So, given that the principal aim of CPGs is to establish a SOC, CPGs are employed to set up the legal SOC in the case of dispute.

In the application of CPGs, there is the effective probability that a court recognizes them as general practice in the medical profession.

A veterinarian who exercised the medical profession in accordance with a CPG would be safeguarded from responsibility in the same manner as one who could show that he/she pursued the ordinary practice. Indeed, if a veterinarian can establish that the course of treatment undertaken, whether or not it led to the patient’s injury or death, was in compliance with the customary practice rightly accepted as proper by a body of skilled and experienced veterinarian practitioners, the veterinarian will be presumed to have acted reasonably given the information at the time.

Vice versa, the failure to observe a specific guideline could lead to deduction that he/she did not operate in accordance with the appropriate rules. Anyway, it would constrain the veterinarian to clarify why the CPG was not applied, for example, because the animal owner would decline to apply the CPGs.

It is important to underline that some CPGs are drawn up through a consent, in which the proposals are a combination of current professional opinions and practice.

Other CPGs are created through a more objective, evidence-based medicine conform procedures, in which the basis is an accurate evaluation of observational data rather than the opinions of veterinarians.

If accepted as proof in veterinary malpractice and liability cases, the guidelines developed in the first way described above would mostly play the identical role as a medical expert witness to ordinary practice. If the courts adopt the ordinary practice to establish the legal standard for good practice, evidence-based medicine obtained from the CPGs will be inadequate or have doubtful applicability in the legal system. 

The pursuit to a proper CPG could be considered by the court as indication of a veterinarian’s “reasonable prudence,” even if it was not confirmed that a significant number of veterinarians had followed the guidelines in clinical practice. 

For the admission by the court of the CPG as the legal standards, the official adoption of the CPG by medical profession would be necessary. As in the case of the minority doctrine, the adoption would be influenced by the significance of the organism(s) establishing, approving, and/or endorsing the CPG.

## 5. Legal Implications

Given that CPGs already used in human medicine may limit physician autonomy and impose inflexible or unrealistic standards on clinical practice [66] and that the veterinarians themselves are possibly resistant to the implementation of CPGs, in our opinion, it is necessary to highlight some legal implications in the development of CPGs in veterinary medicine.

It is primarily important to note that the veterinarian’s duty is also defined by the veterinary profession itself. The veterinarian should act like a skilled and diligent veterinary practitioner in accordance with the current veterinary knowledge and clinical experience. The medical professional standard is established by the veterinary profession (such as ethical code) and not established by the individual veterinarian.

Having said that, we believe that the purpose, the intended use, and the applicability of CPGs should be clearly described in order to affect their subsequent legal standing. They should have practicality and application flexibility. In other words, they should provide a clear indication of practical behavior that can always be modelled in the absolute singularity that characterizes each individual clinical condition.

The first principle that must inspire the formulation of the CPGs is their validity.

Guideline developers should clearly state that CPGs are always voluntary and that they do not define the approach to every individual case. They should also indicate that acceptable medical practice includes a variety of responses to clinical problems. 

If legal prescriptions for guidelines are followed, it is likely that the conduct of malpractice litigation will change somewhat. Experts will use guidelines to support their positions on behalf of either plaintiffs or defendants.

## 6. Conclusions

CPGs play an effective role in malpractice litigation, simplifying decision-making and reducing errors. In fact, veterinarians often welcome such guidelines and use them as a way of improving care, reducing stress and uncertainty, and justifying their practices to clients [67].

It is good public policy to have laws that reflect the changing nature of the relationship between people and their animals, but such laws must consider the impact on the health of animals overall. A balanced approach that provides for capped non-economic damages is an important starting point in recognizing the changed status of companion animals in our society while also encouraging professionals to continue providing quality veterinary care.

## Figures and Tables

**Table 1 animals-09-00577-t001:** List of the most common Guidelines in Veterinary Medicine.

Guidelines in Small Animals Practice	Society/Group Producing	References
Guidelines for feline and canine vaccinations	AVMA, AAHA, and AAFP	[48,49]
End-of-Life Care Guidelines	AAHA/IAAHPC	[50]
Senior Care Guidelines	AAFP	[51]
Feline-Friendly Handling Guidelines	AAFP/ISFM	[52]
Feline Life Stage Guidelines;	AAFP/AAHA	[53]
Anesthesia Guidelines for Dogs and Cats;	AAHA	[54]
Dental Care Guidelines for Dogs and Cats	AAHA	[55]
Canine Life Stage Guidelines;	AAHA	[50]
Consensus Statement: Guidelines for the Diagnosis and Management of Canine Chronic Valvular Disease;	ACVIM	[56]
Nutritional Assessment Guidelines for Dogs and Cats;	AAHA	[57]
Antimicrobial Use Guidelines for Treatment of Urinary Tract Disease in Dogs and Cat	ISCAID	[58]
Guidelines for the Euthanasia of Animal	AVMA	[59]
Consensus Statement: Enteropathogenic Bacteria in Dogs and Cats	ACVIM	[60]
Consensus statement: Guidelines for the identification, evaluation, and management of systemic hypertension in dogs and cats	ACVIM	[61]
Consensus statement: Support for rational administration of gastrointestinal protectants to dogs and cats	ACVIM	[62]

American Veterinary Medical Association (AVMA); American Animal Hospital Association (AAHA) and American Association of Feline Practitioners (AAFP); International Association for Animal Hospice and Palliative Care (IAAHPC); International Society of Feline Medicine (ISFM); American College of Veterinary Internal Medicine (ACVIM); International Society for Companion Animal Infectious Diseases (ISCAID).

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
