# Peer review of "Clinical Practice Guidelines: An Opinion of the Legal Implication to Veterinary Medicine"

_animals, 2019, doi:10.3390/ani9080577_

Round 1

Reviewer 1 Report

Review:

Clinical Practice Guidelines: Legal Implications in Veterinary Medicine

I think the topic is highly relevant. Unfortunately, the authors remain rather vague for many aspects. I am missing a clear description and  a number of relevant details.

Comments:

-simple summary/abstract: in line 15 „considers more carefully damages… liability cases“ – in the remaining part of the paper there is no evidence that there is an increase in liability cases. Please provide some citations or numbers of legal cases illustrating the stated increased consideration of vet malpractice and liability cases.

-line 23: please give more evidence for the statement that „this in turn has led the legal system to assess…“ – later in line 56 it is mentionned thet the legal system more carefully should…“ – but I am missing the explanation/demonstration that the legal system takes damages  in vet practice more serious.

-line 39-42: Presumably this holds for dogs and cats, but not for pigs and poultry. I would suggest that you specify for which species you intend the cpgs to play a legal role? (If you also include veterinary practice for lifestock, then vet practice with regard to food safety should be considered)

-lines 43-45 Some of the references are quite old and (at first glance – I haven’t read them) I got the impression that they are focussing on describing veterinary advandecements – similar to human medicine. I am missing citations who reflect on the similarities / differences between human and veterinary medicine

-line 48. As the link to legal aspects is stressed throughout the paper, please provide more details on how the role of animals has changed (e.g. in Switzerland since 2002 in the Zivilgesetzbuch „Tiere keine Sachen mehr“ (Art. 641a Abs. 1 ZGB), Article 285 of the Austrian Civil Code, which became operational in 1988, provides that ‘animals are not objects; they are protected by special laws’, German Civil Code, § 90a or in Portugal just recently https://www.theportugalnews.com/news/animals-no-longer-considered-things-from-may/41295“). Please give more details on the legal position of animals and the recent changes (at least in the countries from the authors) and explain how this related to the CPG/SOC.

-line 59 Please provide a citation, particularly of CPG which have been implemented (or attempts thereof).

-line 61/62 I think there is a wealth of literature and on-going discussions in human medicine about CPGs. Please reflect on the pros and cons in human medicine and how they apply in vet medicine, similar or different problems?

-line 69, presumably strenghtening the bond between humans and dogs/cats (as in the Special Eurbarometer (2005) being asked: „have you ever visited a farm which rears animals?“ 31% respondents reply „never“ and 13 „Once“. Please describe clearly for which species you think the CPGs should apply? And how this then will affect the different vet specialisations?

-line 90 could you describe more in detail which different principles apply?

-line 98 I think it is crucial also to discuss more in detail who has the role/responsability to „promulgate“ such CPGs. What happens if they contradict each other. For Europe the American CPGs might provide some informations, but I think they cannot be binding as pathogens and vaccines differ.

-line 109 I think you mention here a very relevant point, please expnad and give some citations/cases showcasing the stated mismatch

-line 125 I would suggest that you mention SOCs earlier (i.e. abstract / introduction)

-line 150 does this apply to veterinary medicine, too?

-line 152 could you provide cases (or numbers how many times vets have been sued for these aspects?)

-line 181 please provide more details (when, where, which CPG was considered to be relevant, who provided the CPG)

-line 199 please avoid the usage of the word „correlation“ as this wording does not necessarily imply causality. Presumably you mean here that there is a clear causal association between the action of the vet which led to the damage.

-lines 220 please give examples

-lines 239 Please expand on this – that the vets themselves are possibly resistant to the implementation of CPGs

-line 256/7 Has this happened in human medicine?

Author Response

Dear Reviewer,

thank you very much for your comments.

We have revised the manuscript in accordance to your requests. 

See the upload pdf file in attachment.

Reviewer 2 Report

An interesting reflection on a point of concerns in veterinary practice

Author Response

Dear Reviewer,

We would like to sincerely thank you for your  comment. 

Reviewer 3 Report

This manuscript is not a typical research based paper.  This is more of an opinion-based manuscript with little support data.  The authors do not cite court decisions nor specific legislation.  As an opinion manuscript, this reviewer found it challenging to accept the breadth of statements made without specifying specific jurisdictions (countries).  Canadian law is different than Australian law and is different than the EU agreements.  Negligence can be governed by either statute or common law principles, depending on the system of law.  Nowhere do the authors make this clarification to the readers. 

Line 49 uses the term animals with disregard to specifically farm animals or companion animals.  This reviewer would be hard pressed to find a pork producer with grower pigs that considers these animals as “members of the family or children or best friends…”  The authors should specifically limit their discussion to “companion animals that are brought into a home environment.”  The authors indiscriminately use the term, “animals,” which is too broad.

Line 58 language issue, “did (not) apply (the standard of care).”

Line 63-64, the authors use the term “Civil and penal courts…”Again there are likely jurisdictional differences.  For example, the USA, uses state licensing boards (not penal) to discipline veterinarians who breach the standard of care.  Also, owners in the USA can use civil law via a tort action to make a claim of negligence against a veterinarian for the breach of the standard of care.  This article does not make sense as the authors are trying to cover too much territory (too many different jurisdictional boundaries). 

In lines 101, the authors only cite American organizations without actually using their full name. 

Lines 108 and 109, the authors provide no citation or support for their statement.

The authors fail to include a discussion about the potential for emergency exemptions. 

Lines 150-151, does not make sense.  If a result of legislation, then in the English based law countries legislation = law.

Lines 183-192, the authors fail to use appropriate English language to make the sword and shield argument.  This reviewer suggests that the authors seek out someone who can assist with their English language edits. 

After line 216, a logical addition would be the decline by the owner to apply the CPG and the effect to the practice of veterinary medicine. 

Lines 230-233, very confusing and not sure what the authors are trying to say. 

Lines 242-243, veterinarian’s duty is also imposed by law (not just the veterinary profession itself).  There are licensing requirements in specific countries. 

Following line 251 a statement such as “This is an aspirational goal that may not be achievable in today’s environment of scientific advances.”  Essentially guidelines will be outdated when new procedures, vaccines, drugs or drug use for a clinical syndrome are approved or published in a peer-reviewed journal. 

This reviewer appreciates what the authors are attempting but it does not work.  The authors need to state this is an opinion piece, and the opinion is based on the situation in the authors’ home countries.  They do not have enough supportive data or citations to go beyond that level. 

Author Response

Dear Reviewer, 

thank you for your comments and suggestions. Please see the attachment.

Reviewer 4 Report

Comments and Suggestions for Authors

Line 54: delete underscore

I suggest that in the paragraph 3 “Overview of Clinical Practice Guidelines (CPGs)” you could add (also in a table) a list of the most common Guidelines in Veterinary Medicine together with the references (for example, Guidelines for feline and canine vaccinations produced by the AVMA, AAHA, and AAFP;  AAFP-AAHA Feline Life Stage Guidelines; AAFP 2008 Senior Care Guidelines; AAFP and ISFM Feline-Friendly Handling Guidelines; AAHA Anesthesia Guidelines for Dogs and Cats; AAHA Canine Life Stage Guidelines; ACVIM Consensus Statement: Guidelines for the Diagnosis and Management of Canine Chronic Valvular Disease; AAHA Nutritional Assessment Guidelines for Dogs and Cats; Antimicrobial Use Guidelines for Treatment of Urinary Tract Disease in Dogs and Cats; AVMA Guidelines for the Euthanasia of Animal; ACVIM Consensus Statement: Enteropathogenic Bacteria in Dogs and Cats: Diagnosis, Epidemiology, Treatment, and Control, etc)

Check the references: not all of these are reported according to the guidelines of the Journal

Author Response

Dear Reviewer,

we would like to sincerely thank you for your advices and constructive comments. 

Round 2

Reviewer 1 Report

All my comments in the previous review have been dealt with appropriately. Please check the English (i.e. humane medicine into human medicine)

Author Response

Dear Reviewer 1,

thank you very much for your comments.

The text has been proofread and corrections have been made.

Best regards,

Annamaria

Reviewer 3 Report

The title needs to clearly indicate that this is an opinion paper.  Title suggestion:

Clinical Practice Guidelines:  An Opinion of the Legal Implications to Veterinary Medicine.

Someone needs to go through and carefully review the English text as there are still some word usage problems.  Table 1 has a mix of non-italic and italic font at the bottom.  Lines 282-283 need citations for those lines.  What courts and where?

Lines 314-319 needs a statement that an expert will still need to state what CPG should be used. 

Authors did do significant refinement and improvement of the ms.  This work is still primarily an opinion piece and needs to clearly state that.

Author Response

Dear Reviewer 3,

Thank you for your suggestions.

We agree with your comments, and we corrected point by point the manuscript accordingly as follows:

The title needs to clearly indicate that this is an opinion paper.  Title suggestion: Clinical Practice Guidelines:  An Opinion of the Legal Implications to Veterinary Medicine.

The title was changed as suggested.

Someone needs to go through and carefully review the English text as there are still some word usage problems. 

The text has been proofread and corrections have been made (All changes that we made are highlighted by blue colour for you to see them.)

Table 1 has a mix of non-italic and italic font at the bottom. 

We have corrected the font used in Table 1.

Lines 282-283 need citations for those lines.  What courts and where?

The text presents the opinion of the authors with the aim to underline that whenever an injury results from an activity at the veterinarian’s office that is not considered a rendering of professional veterinary medical services, the cause of action is not malpractice, but ordinary negligence.

It is important to point out that a “duty” is an obligation to satisfy a standard of conduct toward another. One must act reasonably in light of apparent risk. Veterinarians owe to their clients the duty to practice veterinary medicine in such a manner so as to meet the standards expected of the profession. Veterinarians owe this duty to their animal patients and also to their human clients.

Lines 314-319 needs a statement that an expert will still need to state what CPG should be used. 

In the text, we have explained that the main issues commonly calling for the expert testimony are: (1) the applicable standard of care, (2) causation, (3) the assessment of damages, which often involves (4) medical prognosis.

The type of issue involved dictates the nature and scope of the expert witness’s input. The last three of these issues - causation, assessment of damages, and prognosis - usually require the expert witness to apply his or her expertise directly to the question at hand.

Even where a CPG contains information relevant to causation, assessment of damages, or prognosis, it is hard to see how a court could use this information without the accompanying testimony of a medical expert witness.

Authors did do significant refinement and improvement of the ms. This work is still primarily an opinion piece and needs to clearly state that.

Thank you. As suggested, we changed the title to clearly state that the paper presents opinions of the authors.

Best regards,

Annamaria

This manuscript is a resubmission of an earlier submission. The following is a list of the peer review reports and author responses from that submission.